# Accelerating the Diagnosis of Pandemic Infection Based on Rapid Sampling Algorithm for Fast-Response Breath Gas Analyzers

**DOI:** 10.3390/s24196164

**Published:** 2024-09-24

**Authors:** Artur Prokopiuk, Jacek Wojtas

**Affiliations:** Institute of Optoelectronics, Military University of Technology, 2 Kaliskiego Str., 00-908 Warsaw, Poland

**Keywords:** breath sampling, breath analysis, capnography, EVOC, CO_2_ detection, SARS-CoV-2, absorption spectroscopy, gas sensors

## Abstract

This paper presents a novel technique for extracting the alveolar part of human breath. Gas exchange occurs between blood and inhaled air in the alveoli, which is helpful in medical diagnostics based on breath analysis. Consequently, the alveolar portion of the exhaled air contains specific concentrations of endogenous EVOC (exogenous volatile organic compound), which, among other factors, depend on the person’s health condition. As this part of the breath enables the screening for diseases, accurate sample collection for testing is crucial. Inaccurate sampling can significantly alter the composition of the specimen, alter the concentration of EVOC (biomarkers) and adversely affect the diagnosis. Furthermore, the volume of alveolar air is minimal (usually <350 mL), especially in the case of people affected by respiratory system problems. For these reasons, precise sampling is a key factor in the effectiveness of medical diagnostic systems. A new technique ensuring high accuracy and repeatability is presented in the article. It is based on analyzing the changes in carbon dioxide concentration in human breath using a fast and compensated non-dispersive infrared (NDIR) sensor and the simple moving adjacent average (SMAA) algorithm. Research has shown that this method accurately identifies exhalation phases with an uncertainty as low as 20 ms. This provides around 350 ms of breath duration for carrying out additional stages of the diagnostic process using various types of analyzers.

## 1. Introduction

Exhaled air composition analysis is a rapidly developing, non-invasive clinical diagnostics technique. It enables the screening of patients to determine their health condition. Among the endogenous gases in human breath, there are particular gases, so-called biomarkers [1], whose concentration change is characteristic of a given disease. Low concentrations of biomarkers (at the ppb-ppm level [2]) impose specific requirements not only on their detection systems (gas analyzers) but also on the procedures and tools used to collect and prepare the breath sample. For this reason, the technology for sampling human exhaled air is essential and constitutes a separate research issue in developing exhaled air analysis systems [3]. A key challenge for such a system is developing separation procedures that would allow accurate extraction of the volume of exhaled air containing a specific biomarker (or set of biomarkers) from a defined part of the respiratory apparatus undisturbed [4]. The presence of biomarkers differs between the respective parts of the human respiratory system; for example, a high concentration of nitric oxide (NO) can be found in the upper respiratory tract, but its origin is unknown and could be caused by the patient’s addiction to smoking [5]. The NO from the lower respiratory tract carries information about the body’s metabolic processes [6]. The low concentration of valid biomarkers in this part of exhaled air also increases the requirements for the sensitivity of the analyzers in the case of gases such as NO. The most sensitive detection methods are already used, such as laser absorption spectroscopy or chemiluminescence [7,8].

The lower the concentration of an analyzed biomarker of the diagnosed disease, the greater the importance of accurately determining the respiratory phases and their precise separation. At rest, approximately 10–15 breaths per minute are usually taken, and each breath contains 500–600 mL of exhaled air. According to ATS/ERS guidelines [9], the patient should be able to exhale for 10 s with a standard flow of 50 mL/s, where 7 s is air from the lower respiratory tract, the so-called third phase (III) of exhalation. Assuming that the uncertainty of volume of the third phase separation is as low as 5%, the maximum time allocated to executing a separation procedure by a system should be less than 350 ms. Accurate sample collection is essential for rapid testing of large numbers of potential patients, as demonstrated, for example, by the coronavirus 2019 (COVID-19) pandemic. Handling of respiratory samples in the reported studies on rapid COVID-19 diagnostic methods is usually based on Tedlar bags [10] and specific time-consuming procedures related to sample preparation, such as heating and sending. It significantly limits the possibilities of large-scale diagnostics, which are required for many cases.

This article analyses the solutions used so far. It presents a new and fast NDIR-based technique to extract exhaled air samples, ensuring direct delivery of an accurately selected breath sample, enabling the determination of its origin from the human respiratory system while reducing disturbances in biomarker concentrations and providing repeatability of the entire diagnostic system. Aspects of using NDIR sensors for this application are discussed in [11]. The ability to directly transfer an air sample to the fast-response exhalation gas analyzer (e.g., based on laser techniques [12]) provides a valuable opportunity to observe concentration changes during exhalation in a specific phase, rather than measuring the average concentration value, as is the case when a sample is collected into a container (Tedlar bags). The measurement is made after the entire volume of the air sample has been collected.

## 2. Materials and Methods

Breath samples are extracted using many methods, e.g., based on sound analysis of air flow, volume/pressure of exhaled air analysis, or by observing the concentration changes of specific gases such as N_2_ or CO_2_ (Table 1). 

The airflow sound analysis on the respiratory tract is very convenient because it can be applied while the patient is asleep. Varying volumes of collected breath samples or inconsistent duration of sample acquisition and breath part determined by the diagnostician may be sufficient for analyzers that do not require large sample volumes (e.g., GC/MS). More accurate techniques for separating exhalation phases are based on analyzing exhaled air parameters such as flow, pressure, or concentration. The simplest of these uses a fixed threshold value for a specific gas concentration, but its main disadvantage is that it varies from person to person. 

The existing methods for analyzing human exhalation have not yet been standardized [19] and are based on general recommendations [8]. In addition, the reported sampling techniques (acoustic, volumetric, time) are difficult to compare due to the different methodologies used depending on the target of diagnostic procedures. There are also no standard criteria for determining exhalation samples for testing [13]. 

The analysis of the disadvantages of the described methods shows that the concentrations of biomarkers also depend on the patient’s individual characteristics (age, sex, diet, smoking), so identical and precisely defined samples should be provided for their accurate analysis. It can be pretty challenging because the breath sampling methods depend highly on patient cooperation and do not ensure desirable control over the patient’s breathing process [20]. Considering the difficulties with objective and precise separation of breath phases, attention has been focused on the possibility of utilizing changes in the concentration of selected gases in exhaled air. 

Further analysis showed that the analysis of the concentration of nitrogen or carbon dioxide in exhalation provides the most significant possibility of precise extraction of samples from human breath. Nitrogen is used in single-breath nitrogen washout/multiple-breath nitrogen washout (SBNW/MBNW) methods but requires expensive apparatus or indirect measurements of the N_2_ fraction [21]. For this reason, the CO_2_ analysis is vastly superior. Fast and accurate CO_2_ sensors that utilize non-dispersive infrared (NDIR) technology are widely used in healthcare [22]. Typical absorption spectra for the CO_2_ and H_2_O mixture, in the range of 1 μm–11 μm, are shown in Figure 1. 

CO_2_ sensors that use the phenomenon of infrared radiation absorption at a wavelength of 4.3 μm are highly resistant to interference with a gas, including water vapor. Analysis was carried out using HITRAN (version 4.1) software for a 2 cm absorption cell and gas concentrations of CO_2_ (5%) and H_2_O (5.8%). Carbon dioxide and water vapor concentration corresponded to the typical values in exhaled human air. The following simplified relationship exists between water vapor concentration and relative humidity (1):(1)$$RH=\frac{C_{H_{2}O}\cdot P}{sat(T)}$$
where $C_{H_{2}O}$ is HO_2_ concentration, *P* is air pressure and *sat (T)* is the saturation vapor pressure of water at a given temperature *T* (the maximum amount of water vapor the air can hold). The error of RH calculation is less than 0.5% in the temperature range of −50 °C to 50 °C [23]. 

The process can be divided into four phases based on monitoring CO_2_ concentration during exhalation (capnography) (Table 2). Different phases may be interesting for separate research purposes depending on the biomarker. Biomarkers can be found in both the upper respiratory tract: for example, albumin as a potential biomarker of plasma leaking into the airways [24]; VOCs from the oral cavity as a biomarker of oral squamous cell carcinoma [25] and the lower respiratory tract; for example, isoprene, which is a potential biomarker of liver fibrosis [26]; procalcitonin, which appears to be a biomarker of ventilator-associated pneumonia [27]; and nitric oxide as a biomarker of asthma, angina, hyperbilirubinemia [28]. The origins of some of these biomarkers are still not fully elucidated [29], but it seems that the most valuable part of breath for the analysis of VOCs is the third phase [30]. The gases exhaled during this phase come from the lower respiratory tract, which is involved in gas exchange from metabolic processes in the alveoli, where blood-gas exchange occurs [31].

During the second phase an almost linear increase in CO_2_ concentration and flattening of the characteristics specific phase rdcan be observed. However, the beginning of the third phase is not precisely defined (Table 3).

The authors in [38] provided a very useful definition sufficiently based on scientific scrutiny of the beginning of the third breath phase. Based on volumetric capnography, linear regression from the beginning of the second phase and 40% of expired air volume were labelled as expiratory slope function. The alveolar plateau (the third phase) was compared using linear regression of data points between 40% and 80% of exhaled air volume. However, recent reviews [40] observed ambiguity in publications, even in methodological terms, and we therefore propose guidelines to specify the gas exchange measurement or estimations algorithm. The methods presented are often impractical for real-time measurements and the automatic analysis of many samples. The beginning of phase III is only calculated after it ceases.

Additionally, practical characteristics may differ from typical ones (shapes, slopes, noise, fluctuations) [41], which could increase uncertainty and lower repeatability. In this case, a method that minimizes the influence of patient and device-based parameters should be developed. Otherwise, measured biomarker concentrations are considered to be unreliable due to the mixing of air from different phases of exhaled breath.

The authors propose an algorithm that analyses changes in CO_2_ concentration using the SMAA (Simple Moving Adjacent Average) filter, which allows for the precise detection of the beginning of the third breath phase. A constant threshold value of the dSMAA/dt function can be determined corresponding to it. A similar method, called the first derivative wave analysis (FDWA), has helped analyze cardiogenic oscillations with a constant slope of the third phase based on the concentration of exhaled N_2_ and CO_2_ [42]. The proposed method was compared with one presented in [18]. As a procedure for detecting the phase III slope, the authors proposed using the ‘breakpoint’ analysis of the exhaled nitrogen waveform. The breakpoint was determined where the correlation coefficient of the bilinear regression of the data segment of the second–third phase transition reached its maximum value. The analogical shape of the exhaled carbon dioxide waveform can also be helpful for capnography analysis. 

## 3. Developed Breath’s Third Phase Detection Algorithm 

The simplest method to determine the transition point between phases is the analysis of the time-based (dCO_2_/dt) [43] or volume-based (dCO_2_/dv) dynamics [44]. It is straightforward to calculate and implement in autonomic systems. However, it may not be effective for a signal with fluctuations or in any other way deviating from the typical shape of a capnographic waveform obtained from a CO_2_ sensor (Figure 2a). Visible fluctuations of the signal in the third phase indicate the need for an application of additional signal processing technique (Table 4).

In contrast, this solution uses the SMAA algorithm to remove higher frequency fluctuations. SMAA for $n_{x}$ point is calculated as an arithmetic mean average of $n_{x}$ and i chosen symmetrically around the output point (previous and next neighbouring points, from $n_{x-i}$ to $n_{x+i}$), according to Equation (2):(2)$$SMAA=\sum \frac{n_{x-i}+\ldots +n_{x-1}+n_{x}+n_{x+1}+\ldots +n_{x+i}}{2i+1},$$

Additionally, it allows us to calculate the first derivative of SMAA (3)
(3)$$\frac{dSMAA}{dt}=\frac{{SMAA}_{t2}-{SMAA}_{t1}}{t_{2}-t_{1}},$$
where ${SMAA}_{t2}$ and ${SMAA}_{t1}$ denote the moving average of the neighboring points for the time $t_{2}$ and $t_{1}$, respectively. Applying dSMAA/dt allows smoothing of the capnograph data to precisely determine the beginning of the third phase (Figure 2b).

This algorithm was developed using a method based on determining the intersection point of linear regression lines of the second and third phases of exhalation. A constant dSMAA/dt threshold value enables the determination of the beginning of the third phase of exhalation. 

## 4. Experiments

The rapid breath sampling method was studied using the system previously described as the Breath Sampling Unit (BSU) in [3]. As portrayed below, the patient was equipped with a single-use face mask with a breath filter that blocked more than 99.99% of bacteria and viruses in the exhaled stream (Figure 3). 

The appearance of flow in the system indicated the beginning of the patient’s exhalation. If flow occurred, the sample parameters were measured (pressure, humidity, temperature, and CO_2_ concentration). When the first exhalation phase began, the Ev2 solenoid valve opened, sending this part of the exhalation to the gas analyzer. If the patient continued to exhale, the concentration of CO_2_ in the exhaled air started to increase, indicating the beginning of phase II. When the declared dSMAA/dt value (threshold) was exceeded, the phase III solenoid valve (Ev3) opened, allowing this part of the exhaled air to be sent to the analyzer [47]. Separation of the phases of respiration was performed using Lucifer 221S SS316L (Parker, OH, USA) solenoid valves dedicated to medical applications, which have an operating time of approximately 180 ms. Math calculations, I/O procedures, and algorithm realization took less than 5 ms for the ATMega328P microcontroller. For humidity measurements, the SHT21 sensor (Sensirion, Stäfa, Switzerland) was used, while for temperature and pressure monitoring, the BMP180/BMP280 sensors (Bosch Sensortec, Reutlingen, Germany) were implemented. The flow was determined from the pressure drop of the exhaled air using the Hagen–Poiseuille law. The measurement procedure of these four gas sample parameters took less than 165 ms. Consequently, no more than 350 ms was needed to execute the entire algorithm for extracting the selected exhalation phase. The view of the experimental system during the test is presented in Figure 4. 

It enabled achieving a very high repeatability of 95%. The value was calculated based on the variance of the beginning time of the third breath phase of the single user determined by the dSMAA/dt algorithm, concerning this time determined by the analytical method. All data in the system were collected by a control unit and sent to a PC for later processing. 

Practical tests were conducted to verify further the method presented. A dataset of different CO_2_ waveforms was collected from 14 volunteers using a developed technique (Table 5). The selected volunteers considered the different nature of the respiratory system of healthy individuals [48]. The study aimed to experimentally measure the characteristics of CO_2_ changes in exhaled air during the day. Because of this, the volunteers exhaled at different times of the day, before and after exercise, and before and after a meal, which amassed 320 breath samples. 

The CO_2_ measures obtained in this experiment selected readings with fluctuations in CO_2_ concentration during the third phase of exhalation and at different exhalation times. It made it possible to analyze the working of the CO_2_ NDIR sensor under real conditions. The simplified block diagram of the sampling algorithm using Dsmaa/dt algorithm is presented in Figure 5.

Verification was carried out by comparing the results of the Dsmaa/dt algorithm (with estimates of 3 points forward and backward from the analyzed point) with the calculation method of the coefficient of determination (R^2^) presented in [18]. For this purpose, OriginPro software (2024b version) with the Piecewise Fit function was applied. Analysis showed that the beginning of the third breath phase determined by the bilinear regression method indicated the same timestamp as the Dsmaa/dt method with an uncertainty of 20 ms, which led to the conclusion that the proposed algorithm seems reliable (Figure 6). Furthermore, the proposed method is characterized by a more straightforward mathematical description and lower computational complexity.

The authors successfully implemented the Dsmaa/dt algorithm in a single-core 8-bit 16 MHz AVR microcontroller. The device was designed to allow the patient to exhale for as long as possible, providing a collection of as much the third phase sample volume as possible. Parameters of the breath sample: CO_2_ concentration, temperature, pressure, and flow are monitored in real-time. When airflow occurs, the first breath phase begins, and the Dsmaa/dt of CO_2_ concentration can be calculated. When the Dsmaa/dt value threshold is exceeded, the third phase breath sample is directed to the analyzer. The Dsmaa/dt threshold level was experimentally determined in the developed Breath Sampling Unit to a value of 0.85. When the patient stops exhaling, the extraction of the breath sample is complete. For the experimental setup, to obtain 2 L of breath third phase samples, approximately 5–6 exhalations are required, and the entire sample acquisition time is approximately one minute. Therefore, it is a quick method that minimizes inconvenience for the patient. Indeed, this process may be longer in patients with weaker lung function but, pending the experiment, it did not exceed the typical 2 min.

## 5. Results

For the data analysis presented, the CO_2_ NDIR sensor from the UT100C capnograph was used. This sensor was selected because it is intended for medical use and has a fast data readout (20 ms). Many commercially available CO_2_ NDIR sensors are dedicated to process control or safety (where sensor speed is not as essential and readout delay over 1 s is acceptable), but are too slow for exhaled gas concentration analysis. The sensor was calibrated to minimize the impact of air pressure and temperature on CO_2_ readings. Four waveforms were selected from the whole set of breath samples to analyze the results of the dSMAA/dt algorithm that can handle It. They were chosen for their interesting and unusual characteristics, nontypical shapes, and different CO_2_ concentrations. For every case, the beginning of the third breath phase was calculated using linear regression of the expiratory upstroke and the alveolar plateau (Figure 7a–d).

The experimental capnograph waveforms enabled designating the beginning of the third breath phase. Furthermore, the CO_2_ concentration threshold was calculated. It was observed that, for different persons, the threshold of CO_2_ concentration pointing to the third phase was about 4%, but there were cases where it was much higher (more than 4.5%) and sometimes much lower (around 1.7%). Regardless of the differing exhaled CO_2_ concentration, the beginning of the third breath phase was designated, and determining whether the Dsmaa/dt exceeded the threshold value for any of the data in the set did not cause any problems. Moreover, neither different breath duration (between 8 and 12 s) nor capnograph shape affected the ability to designate the third phase. The Dsmaa/dt algorithm seemed insensitive to breath duration, signal fluctuations, and different levels of CO_2_ concentration in exhaled breath.

The verification analysis of the presented algorithm was performed for CO_2_ waveforms characteristic of specific respiratory system diseases. Four highlighted diseases were found in the literature [49,50,51,52]. The waveforms were digitized using WebPlotDigitizer (https://automeris.io/, accessed on 20 May 2024) software and analyzed via algorithm performance. It should be noted that the dSMAA/dt algorithm determined the beginning of the third breath phase respiration in all cases (Figure 7e,f).

A study of the algorithm readings was performed at different exhalation durations to assess the repeatability of the measurements. Waveforms were taken from collected capnograph data with more than 10 s of exhalation time. These exhalations were described as “long”. The second-case waveforms were described as “short”, taking up 2 to 3 s of exhalation time. The beginning of the third phase and the corresponding concentration of CO_2_ were determined (Figure 8).

In both cases, the beginning of the third breath phase was designated. Again, different levels of CO_2_ concentration were observed at the dSMAA/dt threshold. It is noted that for different people, the threshold of CO_2_ concentration pointing to the third phase of the first breath was in the wide range between 2.66% and 4.36%. Different durations of breathing (including rapid ones) did not affect the algorithm’s functionality. 

## 6. Discussion

The developed algorithm was dedicated to determining the beginning of the third breath phase. Its property is the ability to calculate the transition point between the second and third phases. The presented method for assessing the beginning of the third breath phase seems to be useful for precise selection of breath from the upper and lower respiratory tract. The primary usability parameters are presented in the Table 3. By using the dSMAA/dt function, the beginning of the third phase is designated. For the commercially available fast NDIR CO_2_ sensor, a delay of up to 20 ms was observed. It allows for very high repeatability because the differences between individual tests did not exceed 5%. Compared to machine learning algorithms and multi-task techniques [53], the proposed method is relatively simple and easy to implement. Due to the calculations’ low complexity, it seems useful even for low-power devices or those with low computing capabilities. 

This indicates the possibility of using such breath sampling devices and collecting a precise part of human breath. Consequently, it could be helpful in future diagnoses, for example, in the COVID-19 pandemic. Ethyl butanoate, which comes from the lower respiratory tract, was found to be a biomarker of the said disease. Still, the COVID-19 breath test from the upper respiratory tract could prove quicker and more accessible [54]. Furthermore, its early detection of the possibility of predicting disease or difference with early stages of influenza or pulmonary infections has gained much interest [55].

Breath sampling with breath phase selection allows analysis of endogenous gases linked to metabolic activity in the human body and enables scanning for biomarkers in exogenous gases. EVOCs originate from external variables such as diet, drugs, medications, environmental exposure [56], or the activity of microorganisms in the human body [57], which can be biomarker characteristics of specific diseases. 

## 7. Conclusions

A novel algorithm for detecting the beginning of the third phase of human breath was presented. Calculations based on the dSMAA/dt algorithm of CO_2_ concentration in exhaled air can designate a constant threshold corresponding to the beginning of the alveolar breath phase. The presented method was compared with two other techniques. The first was analytical and used linear regression of the expiratory upstroke and alveolar plateau on the capnograph waveform. The second method was based on calculations of the maximum value of the coefficient of determination (R^2^) of bilinear regression of the second and the third breath phases. Both methods obtained the same final results. Based on experimental CO_2_ waveforms from volunteers, the usefulness of the developed dSMAA/dt method was showcased. The beginnings of the third breath phases were designated independently of exhalation time, CO_2_ concentration, and capnographic waveform shape, which proves the adaptative property of the algorithm developed. The research on characteristic exemplary data for specific respiratory disease diseases showed potential for use in clinical practice. Verification tests showed low uncertainty in detecting the desired exhalation phase (20 ms) and achieved very high repeatability (>95%).

Furthermore, due to the current ambiguity in the precise definitions of the breath parts, it could be helpful when standardizing the methods to separate the human breath into separate exhalation phases. A desirable direction for developing these types of devices, which can significantly improve their parameters, is the implementation of artificial intelligence in the algorithm [58].

## Figures and Tables

**Figure 1 sensors-24-06164-f001:**
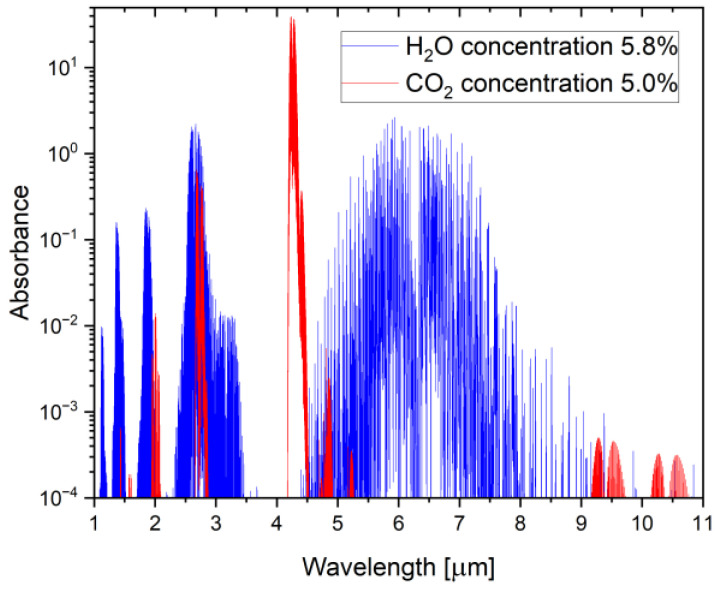
Infrared absorption spectra of CO_2_ and H_2_O, assuming regular concentrations in human breath of 5.0% and 5.8%, respectively.

**Figure 2 sensors-24-06164-f002:**
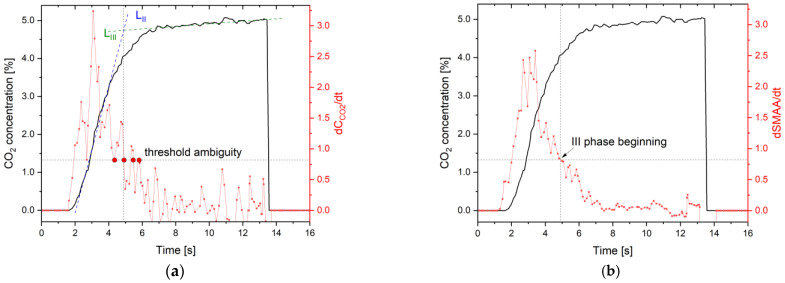
(**a**) An ambitious definition of the beginning of the third phase using the dCO_2_/dt method and an exemplary experimental signal with fluctuations from the NDIR CO_2_ sensor. L_II_ (blue line) is a linear regression of expiratory upstroke, L_III_ (green line) is a linear regression of alveolar plateau, the beginning of the third phase determined by crossing point of L_II_/L_III_ lines, (**b**) dSMAA/dt algorithm and calculation of the start of the third breath phase at the intersection of linear regression slope of L_II_ and L_III_.

**Figure 3 sensors-24-06164-f003:**
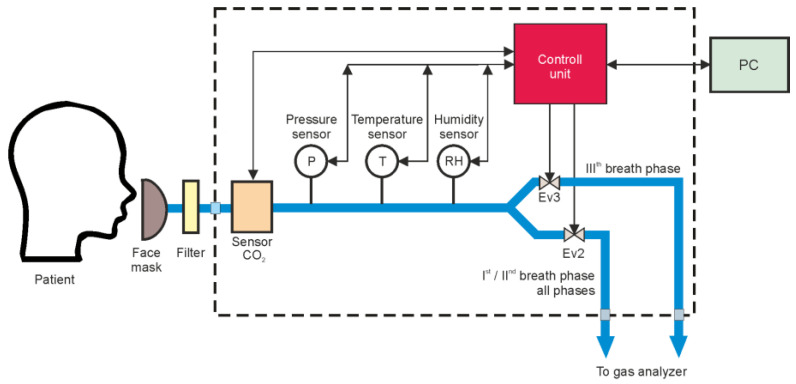
Experimental setup of Breath Sampling Unit.

**Figure 4 sensors-24-06164-f004:**
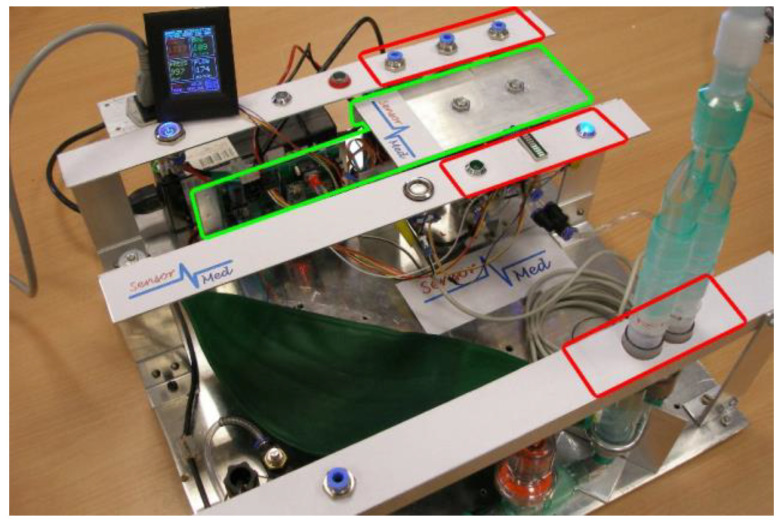
A photo of the experimental system with the implemented algorithm and the required hardware components: control unit (green mark) and patient interface, indicators of the beginning of the patient’s exhalation, and gas analyzer interface (red marks).

**Figure 5 sensors-24-06164-f005:**
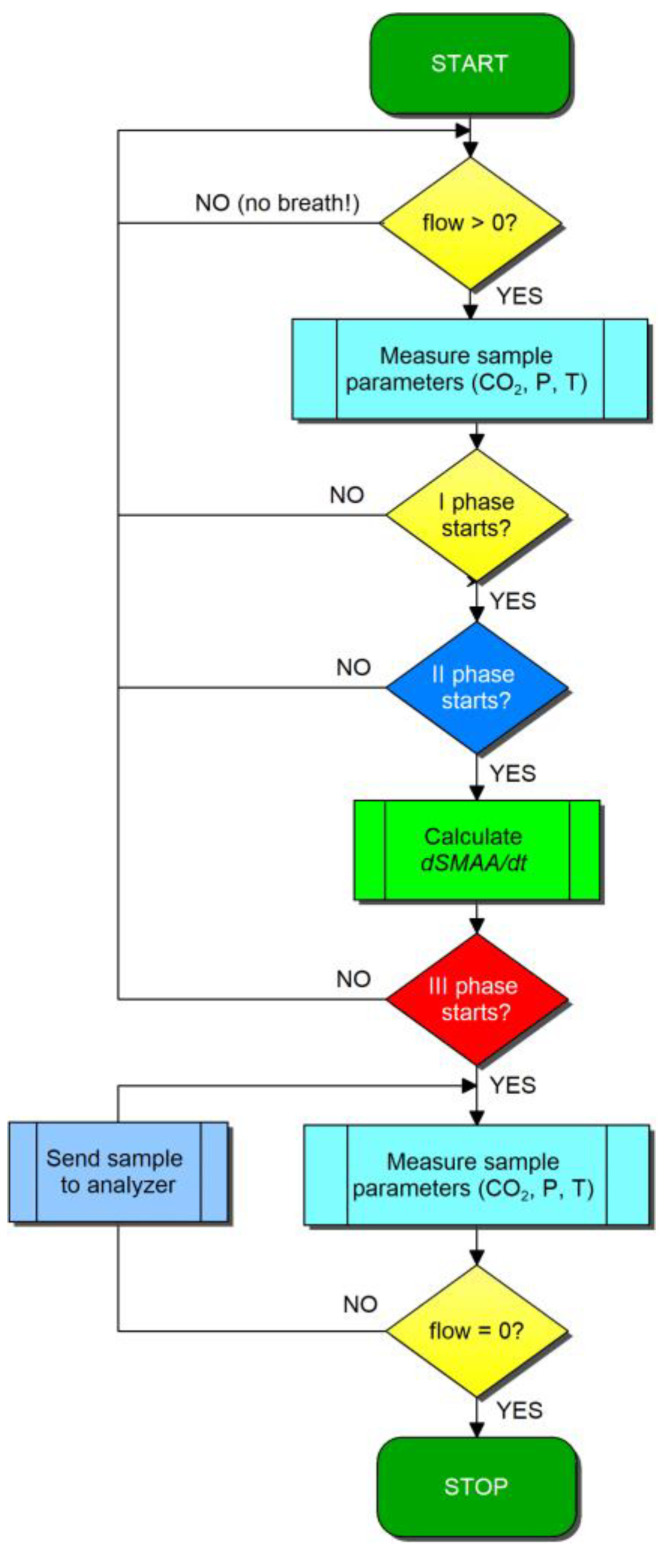
Block diagram of Breath Sampling Unit using the proposed Dsmaa/dt algorithm.

**Figure 6 sensors-24-06164-f006:**
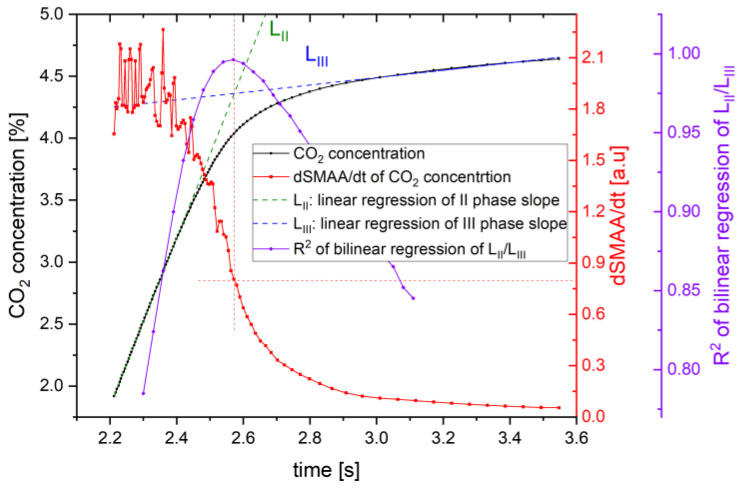
The beginning of breath’s third phase detection with different algorithms. L_II_ (green line) defines the slope of second breath phase, L_III_ (blue line) defines the slope of the third breath phase—the intersection point of LII, and L_III_ defines the start time of the III phase using the analytical method. The same point is designated by calculating the maximum value of the coefficient of determination (R^2^) of L_II_/L_III_ bilinear regression and calculating the threshold value of Dsmaa/dt method.

**Figure 7 sensors-24-06164-f007:**
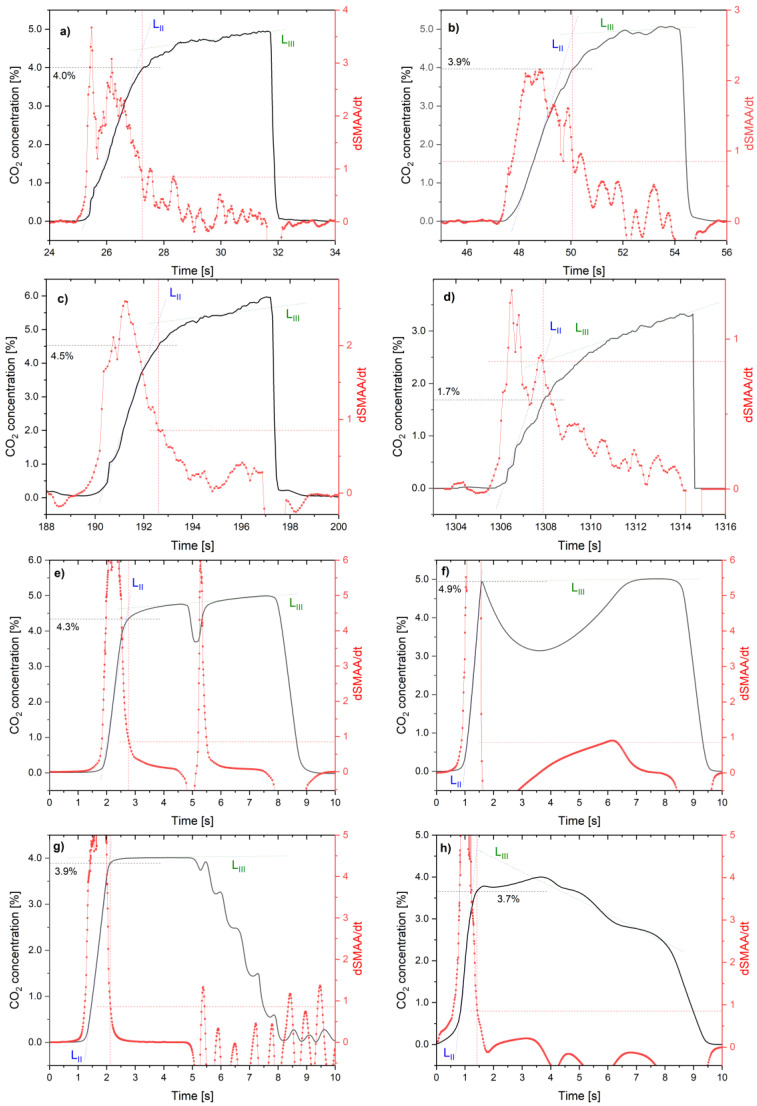
Analysis of the dSMAA/dt algorithm for various experimental capnographic waveform sets: (**a**) close to normal (typical one), (**b**) with the phase III fluctuations, (**c**) with the phase III “boost”, (**d**) “shark fin” shape, (**e**) lung muscle relaxation, (**f**) kyphoscoliosis, (**g**) cardiac oscillations, (**h**) pneumothorax. Black line—raw CO_2_ concentration smoothed with SMAA filtering, L_II_, L_III_—linear regression of expiratory upstroke and alveolar plateau, red—characteristic dSMAA/dt calculated from CO_2_ concentration with designated constant value corresponding to the third phase of the beginning of breath.

**Figure 8 sensors-24-06164-f008:**
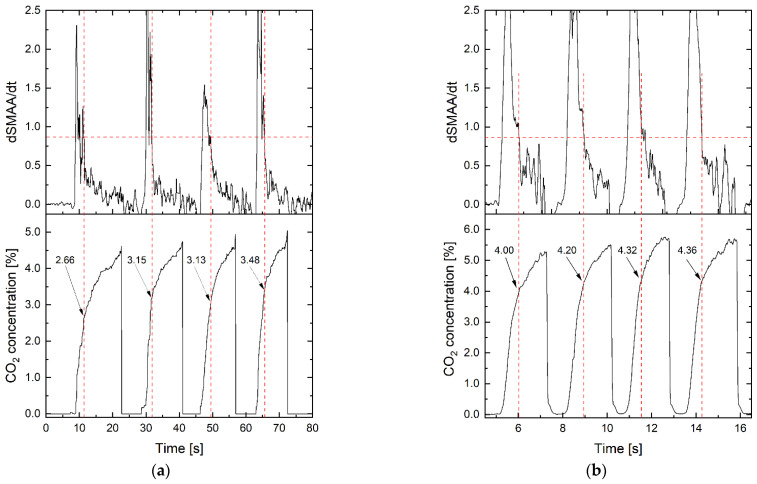
Determining the beginning of the III breath phase and corresponding CO_2_ concentration for (**a**) “long” and (**b**) “short” breaths.

**Table 1 sensors-24-06164-t001:** Review of breath sampling methods for testing.

Principle of Operation	Disadvantages	Literature
Analysis of the sounds of air flowing through the trachea or in the lungs	Relatively complex signal analysis (spectral analysis, phase-shift information, neuro-fuzzy systems)	[13]
Assumed constant volume or duration of phases	The phases’ duration and volumes vary from patient to patient and depend on gender, age, etc.	[14]
Determined by the operator	An experienced operator is required, and identification is subjective	[15]
Analysis of pressure changes in the patient’s exhalation	Identification results depend on many important factors resulting from the patient’s health condition	[16]
Constant threshold of selected gas concentration	The threshold may be different for each patient	[17]
Analysis of dynamics of gas concentration changes	Complex procedures that require a lot of computing power and time, usually performed after exhalation	[18]

**Table 2 sensors-24-06164-t002:** Respiratory phases determined by the CO_2_ concentration in the exhaled air [32,33].

Breath Phase	Description
I	Beginning of exhalation. Air from the mouth and trachea (known as dead space) that is not involved in gas exchange. Low CO_2_ concentration, is usually used as a reference in the analysis.
II	Mixed air from dead space and alveoli. A rapid increase in CO_2_ concentration to approximately 3–5%.
III	Air from the alveoli. Slow increase of the CO_2_ concentration to the maximum level known as the end-tidal point (EtCO_2_).
IV	Very fast decrease in CO_2_ concentration to the ambient level and the beginning of the following breath process.

**Table 3 sensors-24-06164-t003:** Definitions of the beginning of the third breath phase.

Definition	Literature
The point is generally placed between the expiratory stroke and the alveolar plateau.	[34]
The beginning of the flat part of CO_2_ changes as a function of time (time-resolved capnography)	[35]
The beginning of the flat part of CO_2_ changes as a function of volume (volume-resolved capnography)	[36]
The point of intersection of both slopes is determined by the linear regression method.	[37]
Dependencies between specific constant breath volumes	[38]
Constant CO_2_ concentration threshold about 3% ÷ 3.8%	[39]
Analysis of dynamics of gas concentration changes	[16]

**Table 4 sensors-24-06164-t004:** Common methods of capnograph signal processing.

Methods	Literature
Finite impulse response (FIR) 10 Hz low pass filter and moving average filters to limit signal bandwidth, remove noise, and smooth raw CO_2_ signal	[16]
Averaging data for removing fluctuations in ventilation	[45]
Dedicated filtering methods	[46]

**Table 5 sensors-24-06164-t005:** Basic characteristics of volunteers.

Sex (M—Male, F—Female)	Age (Years)	Weight (kg)	Height (cm)	Smokers (Y—yes, N—No)	Number of Air Samples Collected
M	46	85	176	N	30
F	35	60	164	N	40
M	26	63	166	N	50
F	42	52	164	N	30
M	45	98	186	N	30
M	47	70	170	N	30
F	33	58	169	N	40
M	46	77	180	N	30
M	48	87	172	N	10
M	36	76	169	N	10
F	52	55	164	N	10
F	49	58	165	N	10

## Data Availability

All data that support the findings of this study are included within the article.

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
