# Peer review of "Accelerating the Diagnosis of Pandemic Infection Based on Rapid Sampling Algorithm for Fast-Response Breath Gas Analyzers"

_sensors, 2024, doi:10.3390/s24196164_

Round 1

Reviewer 1 Report

Comments and Suggestions for Authors

This paper (sensors-3202502) presents a algorithm for detection of human breath III phase beginning based on NDIR CO2 detection. The techniques used are adequate and the research results are acceptable, but there are many issues that need to be addressed. The manuscript may be accepted after major modifications.

1.       In Introduction: Too much research background is introduced, but there is a lack of relevant research status of NDIR CO2 sensor and exhaled breath detection method, and the same problem also appears in Abstract.

2.       It is mentioned that “regular CO2 (5%) and H2O (5.8%) concentrations in human breath”, but as far as I know, the relative humidity (RH) of human exhaled air is very high, about 80%-90% RH.

3.       It is mentioned that “Rapid increase in CO2 concentration to about 3% ÷ 5%”, is that wrong?

4.       It is mentioned that “the volunteers exhaled at different times of the day, before and after exercise, and before and after a meal which gave 320 breath samples”, how were these sample data used in this paper?

5.       How selective is this detection method for other exhaled gases, such as NH3NOVOCs?

6.       It is mentioned that “For commercially available fast NDIR CO2 sensor a delay of up to 20 ms is observed”, please provide comparative data of relevant commercial sensors.

7.       How to calculate repeatability (5%)?

8.       Most of the cited references are old.

Comments on the Quality of English Language

Moderate editing of English language required.

Author Response

Responses to Reviewer comments have been posted in the file.

Reviewer 2 Report

Comments and Suggestions for Authors

The paper discusses applying IR sensor for a breath analysis via CO2 monitoring where authors proposed to employ primarily a derivative of the signal with time. It is an interesting approach and the results are worth to publish out in Sensors journal. I have only two minor concerns:

1) authors do not mention applying a multisensor arrays based on metal oxide gas sensors for a breath analysis in Introduction and Table 1 that is incorrect, should be improved;

2) authors have to supply some details about volunteers who were involved into study (sex, smokers, etc).

Comments on the Quality of English Language

English is fine.

Author Response

(The authors gave the same response as above.)

Reviewer 3 Report

Comments and Suggestions for Authors

Review Comments

The authors present an interesting use of the Simple Moving Adjacent Average (SMAA) algorithm for detecting breath biomarkers, particularly focusing on real-time detection of the critical phase III in exhalation. While the manuscript has potential, several areas need significant revision to enhance its clarity, depth, and innovation. Therefore, I recommend major revisions before reconsideration. Below are the key points that need to be addressed:

The manuscript contains numerous grammatical issues that hinder readability. For example, in the abstract, the sentence "Therefore, this part of breath enables screening for diseases and accurate sample collection for testing is very important." requires revision. The authors should thoroughly review and correct such language issues throughout the manuscript to improve overall clarity.

While the manuscript touches on the potential of the respiratory system in clinical applications, the authors should place more emphasis on the detailed introduction and analysis of the algorithms and their outputs, specifically regarding disease diagnosis and treatment results. The current focus on gas collection methods is somewhat disproportionate and could be moved to the supplementary material for brevity.

Although the proposed SMAA algorithm is compared with linear regression methods, the authors are encouraged to include a more comprehensive comparison with state-of-the-art machine learning algorithms that are increasingly popular for breath biomarker detection. This will provide a better understanding of how the proposed method performs relative to more advanced approaches.

The manuscript reports that the algorithm performs well on a small-scale dataset. However, to substantiate the robustness and generalizability of the proposed method, further validation using larger clinical datasets is necessary. This would greatly enhance the manuscript’s contribution to the field.

Currently, deep learning models and multi-task techniques are widely used and may be considered for comparison with the traditional models used in the manuscript. There are many studies about the deep learning methods of the electronic nose for VOCs detection. I suggest the authors have some review or discussion of these articles if appropriate.

[1] W. Ni, T. Wang, Y. Wu, X. Liu, Z. Li, R. Yang, K. Zhang, J. Yang, M. Zeng, N. Hu, B. Li, Z. Yang, Multi-task deep learning model for quantitative volatile organic compounds analysis by feature fusion of electronic nose sensing, Sens. Actuators B: Chem. 417 (2024) 136206. https://doi.org/10.1016/j.snb.2024.136206.

By addressing these concerns, the manuscript could be strengthened and better aligned with current research trends in breath biomarker detection.

Comments on the Quality of English Language

The English language needs to be polished and revised. There are also some grammatical, syntactical, and other errors of experssion in the manuscript that may make reading and comprehension difficult.

Author Response

(The authors gave the same response as above.)

Round 2

Reviewer 1 Report

Comments and Suggestions for Authors

1.      In Question 2, to facilitate readers’ understanding, the conversion relationship between “percentage of water vapor” and “relative humidity” should be supplemented in the main text.

2.      In Question 3, I think the connection symbol is incorrect, such as “concentration threshold about 3% ÷ 3,8%” in line 140, “to approximately 3% 5%” in line 134.

3.      In Question 6, to facilitate readers’ understanding, relevant explanations should be added to the main text.

Comments on the Quality of English Language

English writing and paper formatting need to be improved.

Author Response

Responses to the reviewer's comments are provided in the file.

Reviewer 3 Report

Comments and Suggestions for Authors The authors have revised the manuscript and appropriately addressed the questions in the comments.
I recommend accepting the current version of the manuscript.
However, I also suggest the authors to modify some small mistakes in the proof stage,
e.g., the authors list and format of the new-added references.

Author Response

(The authors gave the same response as above.)
